# Development and validation of a predictive nomogram for vascular crises in oral and maxillofacial cancer patients undergoing free flap surgery

**Ying Zheng[1,2‡], Jingya Yu[3‡], Yunyu Zhou[4], Qian Lu[3,5], Yu Zhang[3], Xiaoqin Bi●[1]***

**1** State Key Laboratory of Oral Diseases & National Center for Stomatology & National Clinical Research Center for Oral Diseases & Dep. Of Orthognathic and TMJ Surgery, West China Hospital of Stomatology, Sichuan University, Chengdu, China, **2** Chengdu Sixth People's Hospital, Chengdu, China, **3** West China School of Nursing, Sichuan University, Chengdu, China, **4** School of Stomatology, North Sichuan Medical College, Nanchong, Sichuan, China, **5** Chengdu Fifth People's Hospital, Chengdu, China

‡ YZ and JY are the first co-authors and contributed equally to this work.
* hxbxq@163.com

**Data Availability Statement:** All relevant data are within the manuscript.

## Abstract

### Objective

To develop and validate a predictive model for identifying vascular crises following free tissue flap transplantation in patients undergoing surgery for oral and maxillofacial tumors.

### Methods

This retrospective cohort study utilized medical records from the Department of Head and Neck Oncology, West China Hospital of Stomatology, Sichuan University, covering the period from January 2014 to December 2021. The analysis included 1,786 cases, divided into a training group (n = 1,251) and a validation group (n = 535). Variables included demographic factors, clinical characteristics, and surgical details. Univariate and multivariate logistic regression analyses were performed to identify significant predictors, which were then incorporated into a nomogram. The model's performance was assessed using the concordance index (C-index), receiver operating characteristic (ROC) curve, and decision curve analysis (DCA).

### Results

The incidence of vascular crisis was 5.8% in the training group and 4.9% in the validation group. Significant predictors included tissue flap width, D-dimer levels, preoperative hemoglobin, hemoglobin difference before and after surgery, and type of venous anastomosis. The nomogram showed strong predictive performance with an AUC of 0.780 in the training group and 0.701 in the validation group. Calibration curves indicated excellent fit, and DCA demonstrated clinical applicability.

**Funding:** This work was supported by the Sichuan Science & Technology Project [Grant number 2022JDKP007] and Medical Research Project of Chengdu Health Commission[Grant number 2022015]. Medical research topic in Sichuan Province[Grant number S23099].

**Competing interests:** The authors have declared that no competing interests exist.

## Conclusion

A user-friendly model was developed for detecting vascular crises in oral and maxillofacial tumor patients. This model exhibits robust discriminative ability, precise calibration, high specificity, and significant clinical applicability, effectively identifying high-risk patients prone to vascular crises.

## 1. Introduction

Oral and oropharyngeal cancer rank sixth among prevalent malignancies globally [1–3]. Annually, more than 600,000 individuals are diagnosed with head and neck cancer, with oral cancer accounting for approximately 400,000 of these new cases, exhibiting a notable upward trend in incidence. The 5-year survival rate among individuals afflicted with advanced oral and maxillofacial tumors stands below 40 [4]. A comprehensive investigation conducted at the University of California revealed a disease-specific mortality rate of 23.8%. Contemporary therapeutic approaches for oral and maxillofacial tumors predominantly emphasize surgical interventions complemented by interdisciplinary cooperation.

Despite advancements in surgical techniques, patients undergoing treatment for oral and maxillofacial tumors frequently experience significant postoperative deformities. These deformities can severely impair critical functions such as mastication, deglutition, articulation, and aesthetics, ultimately diminishing the patient's quality of life. FFree tissue flap reconstruction is validated to enhance patients' quality of life and is considered the benchmark for addressing oral and maxillofacial defects [5–9]. Studies report free tissue flap reconstruction success rates for oral and maxillofacial defects range from 88.8% to 100% [10, 11]. However, perioperative vascular compromise remains a leading cause of flap failure and necrosis, posing a significant challenge in clinical practice [12–15]. Currently, there is a lack of prognostic models for vascular complications following free tissue flap reconstruction in patients with oral and maxillofacial tumors. Addressing this gap is crucial, as early identification and intervention can significantly improve clinical outcomes and patient quality of life.

Understanding the factors that contribute to the vascular crisis following free tissue flap transplantation in patients with oral and maxillofacial tumors is crucial for improving surgical outcomes and patient quality of life. However, the current approach does not allow for the objective identification of patients at higher risk for vascular complications. This study aims to develop and validate a comprehensive risk prediction model for vascular crisis in patients undergoing free tissue flap transplantation for oral and maxillofacial tumors.

## 2. Methods

### 2.1 Study setting and participants

This study utilized a retrospective cohort design. Medical records were retrieved for patients who underwent free tissue flap surgery in the head and neck oncology department of West China Hospital of Stomatology, Sichuan University, from January 1, 2014, to December 31, 2021. The inclusion criteria were as follows: (1) Patients diagnosed with oral and maxillofacial tumors (2) underwent free tissue flap repair procedures; (3) operations were conducted immediately following the resection of oral and maxillofacial tumors.

## 2.2 Data collection

During the 8 years from 2014 to 2021, a total of 1,786 free tissue flap surgeries were performed. Of these, 98 cases experienced a vascular crisis, and 63 cases resulted in eventual necrosis, yielding a success rate of 96.5%. The data extraction process was conducted meticulously to ensure accuracy and completeness. All relevant data for the study were extracted from the hospital's electronic medical records (EMR) system using the hospital's EMR data extraction tool. The data were accessed for research purposes between September 1, 2021, to November 30, 2021.

The data were fully anonymized before being accessed by the research team. During and after data collection, none of the researchers had access to any identifying information related to individual participants. Furthermore, given the retrospective nature of the study, it was not feasible to obtain informed consent from individual patients for the use of their data. However, the research protocol was reviewed and approved by the ethics committee, which granted a waiver for the informed consent requirement due to the study's characteristics. All procedures involving human participants were conducted in accordance with the ethical standards of the 1964 Helsinki Declaration and its subsequent amendments or comparable ethical standards. Approval was obtained from the Ethics Committee of West China Hospital of Stomatology, Sichuan University (WCHSIRB-CT-2021-378).

The R software package was used to randomly divide the 1,786 cases into a training group and a validation group in a 7:3 ratio. Consequently, 1,251 cases were included in the training group and 535 cases in the validation group.

**2.2.1 Vascular crisis assessment.** Vascular crisis [16] encompasses venous, arterial, and compound arteriovenous crisis. The occurrence of a vascular crisis was assessed by examining surgical records within the electronic medical record system for evidence of exploration of the anastomosed arteries and veins. These exploration findings were integrated with clinical assessments to provide a comprehensive evaluation of vascular crises in patients undergoing free flap surgery for oral and maxillofacial cancer.

**2.2.2 Model input feature.** Based on previous literature research, this study included a comprehensive set of variables encompassing demographic factors and clinical characteristics. These variables included age, gender, surgery date, complications (hypertension, diabetes, coronary heart disease), tumor site, lifestyle factors (smoking, drinking), preoperative treatments, tissue flap details (type, size), venous anastomosis tools/methods, and blood test results (fibrinogen, albumin, D-dimer, hemoglobin, potassium, prothrombin time, thrombin time, C-reactive protein). A total of 1786 cases were analyzed, with 30 variables assessed. To ensure the integrity of the modeling process, missing data were rigorously examined. Multiple imputations for random missing data were performed using SPSS 26.0 software, while the median value was used to address other missing data points.

## 2.3 Sample size calculation

**2.3.1 Sample size estimation for clinical prediction models.** The sample size for the clinical prediction model was estimated based on candidate prediction parameters, total sample size, and outcome ratio [17], Using eight years of data, the incidence of vascular crises was approximately 5%, with P representing the number of candidate prediction parameters. This study included 30 candidate variables, setting MAPE (mean absolute prediction error) at 0.05. The required sample size for model development was 544 cases, which were all enrolled, while

the training group comprised 1251 cases, meeting study requirements.

$$N = \exp\left(\frac{-0.508 + 0.259\ln(\emptyset) + 0.504\ln(P) - \ln(MAPE)}{0.544}\right)$$

**2.3.2 EPV principle in model stability.** According to EPV guidelines, ensuring at least 10 positive events per model variable stabilizes the model. However, Vittinghoff et al. [18], suggest this guideline is often too conservative. In our binary logistic regression analysis, EPV did not significantly impact confidence interval coverage below a 10% incidence rate. Similarly, with over 30 positive events, EPV ranges of 5–9 and above 10 showed comparable $\alpha$ error rates and biases. In our study's training group, 72 positive events occurred, with a 5% incidence rate, meeting modeling criteria with an EPV greater than 5, ideally involving fewer than 14 model variables.

## 2.4 Statistical analyses

Statistical analyses were conducted using SPSS version 26.0 and R version 4.2.1 software. Normality testing employed the Kolmogorov-Smirnov method. In the training group, single-factor analysis was conducted. Quantitative data with normal distributions underwent the two-independent sample t-test, while non-normally distributed quantitative data were analyzed using the two-independent sample nonparametric test. The Chi-square test was applied for nonparametric analysis of two independent samples, and the rank sum test for dummy variable settings in multicategorical data. Statistical significance was set at $P < 0.05$ (two-sided). The R software package initialized a random seed "1357". The 8-year dataset was randomly split into training and validation groups at a 7:3 ratio [19], followed by a comprehensive comparison of data between the two groups.

## 2.5 Establishment and validation of a nomogram

The development process involved several key steps. Initially, univariate analysis identified significant predictors ($p < 0.05$), which were then integrated into multivariate logistic regression models to construct a predictive nomogram. Each predictor was scored in the nomogram, with higher total scores indicating an increased risk of vascular crisis.

Internal validation utilized 1,000 Bootstrap resamples to confirm model robustness and calibration, with a calibration curve aligning closely with the 45° diagonal line. External validation on the validation group assessed discrimination using the concordance index (C-index) and receiver operating characteristic (ROC) curve, with an area under the curve (AUC) > 0.9 indicating strong predictive performance [20]. Calibration plots compared predicted probabilities against actual outcomes, ensuring reliability, while Decision Curve Analysis (DCA) evaluated clinical utility, highlighting its potential to guide patient care in assessing risk for free tissue flap transplantation.

## 3. Results

### 3.1 Characteristic of patients

A total of 1786 patients underwent free tissue flap surgery, with 1251 in the training group and 535 in the validation group. Table 1 compares the variables between these groups. Crisis incidence was 5.8% in the training group and 4.9% in the validation group. The types of crises (venous, arterial, and both) were similarly distributed across the groups. The mean age was 57.35 ± 13.62 years in the training group and 56.36 ± 12.93 years in the validation group.

**Table 1. Comparison of variables between the training group and validation group.**

| Variables | classification | Training group (n = 1251) | Validation group (n = 535) | $x^2$ /t /z | p |
|---|---|---|---|---|---|
| Crisis | No | 1179 (94.200%) | 509 (95.100%) | 0.580 | 0.446 |
| | Yes | 72 (5.800%) | 26 (4.900%) | | |
| Type of crisis | No crisis | 1179 (94.200%) | 509 (95.100%) | 2.052 | 0.562 |
| | Venous crisis | 43 (3.400%) | 19 (3.600%) | | |
| | Arterial crisis | 23 (1.800%) | 6 (1.100%) | | |
| | Both | 6 (0.500%) | 1 (0.200%) | | |
| Age | | 57.350 ± 13.620 | 56.360 ± 12.930 | 0.818 | 0.414 |
| Fibrinogen concentration | | 3.140 ± 0.900 | 3.120 ± 0.900 | 0.878 | 0.380 |
| Preoperative Alb | | 42.680 ± 3.080 | 42.880 ± 3.120 | -1.116 | 0.265 |
| Preoperative Hb | | 136.710 ± 15.930 | 136.470 ± 16.230 | -0.020 | 0.984 |
| Postoperative Hb | | 113.040 ± 16.820 | 112.53 ± 17.350 | 0.333 | 0.739 |
| PT | | 11.180± 0.870 | 11.210 ± 0.900 | -0.916 | 0.360 |
| Sex | Male | 805 (64.300%) | 340 (63.600%) | 0.103 | 0.748 |
| | Female | 446 (35.700%) | 195 (36.400%) | | |
| Radiotherapy | Without | 1206 (96.400%) | 514 (96.100%) | 0.113 | 0.736 |
| | With | 45 (3.600%) | 21 (3.900%) | | |
| Diabetes | Without | 1116 (89.200%) | 471 (88.000%) | 0.519 | 0.471 |
| | With | 135 (10.800%) | 64 (12.000%) | | |
| History of flap operation | No | 1188 (95.000%) | 512 (95.700%) | 0.444 | 0.505 |
| | Yes | 63 (5.000%) | 23 (4.300%) | | |
| | Yes | 544 (43.500%) | 209 (39.100%) | | |
| Hypertension | Without | 998 (79.800%) | 438 (81.900%) | 1.042 | 0.307 |
| | With | 253 (20.200%) | 97 (18.100%) | | |
| Type of flap | ALTF | 920 (73.500%) | 391 (73.100%) | 0.708 | 0.983 |
| | FFF | 73 (5.800%) | 32 (6.000%) | | |
| | RFFF | 148 (11.800%) | 67 (12.500%) | | |
| | LDFF | 40 (3.200%) | 19 (3.600%) | | |
| | LAFF | 36 (2.900%) | 13 (2.400%) | | |
| | Other | 34 (2.700%) | 13 (2.400%) | | |

**Alb**: Albumin; **Hb:** Hemoglobin; **PT**:Prothrombin time; **ALTF:**Anterolateral thigh flap; **FFF:**Forehead Flap; **RFFF:**Radial Forearm Free Flap; **LDFF:**Latissimus Dorsi Free Flap; **LAFF:**Lateral Arm Free Flap.

Fibrinogen concentration, preoperative and postoperative hemoglobin levels, and prothrombin time showed no significant differences (all p>0.05). In terms of the types of free flaps employed, the Anterolateral Thigh Flap (ALTF) emerged as the predominant choice, representing 73.5% of the training group and 73.1% of the validation group. The Free Fibular Flap (FFF) constituted 5.8% of the training group and 6.0% of the validation group, while the Radial Forearm Free Flap (RFFF) was utilized in 11.8% of the training group and 12.5% of the validation group. In summary, patient characteristics between the training and validation groups were well-matched with no significant differences.

### 3.2 Univariate and multivariate analyses of training set

Table 2 summarizes the univariate analysis of quantitative data for the training group, comparing variables between patients with and without crises. Significant differences were found in

**Table 2. (a) Results of Univariate analysis of quantitative data in training group.** (b) Results of univariate analysis of classified data in training group.

(a)

| Variables | crisis | M(P25,P75) | Z | P |
|---|---|---|---|---|
| Width of flap | without | 4.300(4.000,4.650) | -5.530 | <0.001*** |
| | with | 4.650(4.400,5.000) | | |
| D-dimer | without | 0.440(0.320,0.590) | -3.172 | 0.002** |
| | with | 0.355(0.295,0.440) | | |
| | with | 3.700(3.460,3.910) | | |
| TT | without | 17.600(16.600,18.500) | -2.091 | 0.037* |
| | with | 18.100(17.000,18.800) | | |
| Age | without | 60.000(50.000,67.000) | -2.054 | 0.040* |
| | with | 55.500(45.000,65.000) | | |
| Preoperative Hb | without | 137.000(128.000,147.000) | -2.318 | 0.020* |
| | with | 140.000(133.500,152.000) | | |
| Hb difference before and after surgery | without | 23(15,31) | -2.616 | 0.00** |
| | with | 26(19,36) | | |

(b)

| Variable | classification | Without crisis group (n = 1179) n(%) | Crisis group (n = 72) n(%) | $x^2$/z | p |
|---|---|---|---|---|---|
| Sex | Female | 422(94.600) | 24(5.400) | 0.179 | 0.672 |
| | Male | 757(94.000) | 48(6.000) | | |
| Diabetes | Without | 1050(94.100) | 66(5.900) | 0.479 | 0.489 |
| | With | 129(95.600) | 6(4.400) | | |
| Drink | No | 665(94.100) | 42(5.900) | 0.103 | 0.748 |
| | Yes | 514(94.500) | 30(5.500) | | |
| Hypertension | Without | 939(94.100) | 59(5.900) | 0.223 | 0.637 |
| | With | 240(94.900) | 13(5.100) | | |
| Age of surgeon | 1960s | 120(93.000) | 9(7.000) | -0.629 | 0.530 |
| | 1970s | 668(94.200) | 41(5.800) | | |
| | 1980s | 391(94.700) | 22(5.300) | | |
| Year of Surgery | 2014 | 24(82.800) | 5(17.200) | -3.333 | 0.001*** |
| | 2015 | 52(83.900) | 10(16.100) | | |
| | 2016 | 72(92.300) | 6(7.700) | | |
| | 2017 | 123(93.900) | 8(6.100) | | |
| | 2018 | 174(96.100) | 7(3.900) | | |
| | 2019 | 198(93.400) | 14(6.600) | | |
| | 2020 | 233(95.500) | 11(4.500) | | |
| | 2021 | 303(96.500) | 11(3.500) | | |

TT:Throm bintime; Note

*express p<0.05

** express p<0.01

*** express p<0.001.

several parameters, including tissue flap width, D-dimer, thrombin time, age, hemoglobin difference before and after surgery, preoperative hemoglobin, surgical year, and venous anastomosis tool. The detailed results are presented in Table 2.

**Table 3. Multiple logistic regression analysis.**

| Variables | classification | *β* | *SE* | *wald* | *P* | OR(95%CI) |
|---|---|---|---|---|---|---|
| preoperative and postoperative Hb difference | | 0.026 | 0.010 | 6.055 | 0.014* | 1.026(1.005~1.047) |
| Preoperative Hb | | 0.013 | 0.009 | 2.097 | 0.148 | 1.013(0.995~1.032) |
| Anastomosis tool of veins | Manual | | | 10.362 | 0.006** | |
| | Stapler | -0.783 | 0.285 | 7.544 | 0.006** | 0.457(0.261~0.799) |
| | Both | -1.494 | 0.635 | 5.529 | 0.019* | 0.225(0.065~0.780) |
| Width of flap | | 1.061 | 0.197 | 29.134 | <0.001*** | 2.889(1.965~4.246) |
| D-dimer | | -2.149 | 0.597 | 12.933 | <0.001*** | 0.117(0.036~0.376) |
| Preoperative Alb | | -0.113 | 0.046 | 5.980 | 0.014* | 0.893(0.816~0.978) |
| PT | | -0.067 | 0.161 | 0.174 | 0.676 | 0.935(0.682~1.282) |
| History of flap operation | | 0.804 | 0.485 | 2.740 | 0.098 | 2.234(0.863~5.784) |
| Defect sites | Tongue | | | 6.589 | 0.361 | |
| | Cheek | -0.016 | 0.317 | 0.003 | 0.960 | 0.984(0.529~1.832) |
| | Mouth floor | -0.173 | 0.493 | 0.124 | 0.725 | 0.841(0.320~2.210) |
| | Mandible | -0.904 | 0.576 | 2.463 | 0.117 | 0.405(0.131~1.252) |
| | Upper gum | 0.089 | 0.672 | 0.018 | 0.894 | 1.093(0.293~4.080) |
| | Palate | -1.054 | 1.057 | 0.995 | 0.319 | 0.349(0.044~2.765) |
| | Other | -1.089 | 0.585 | 3.468 | 0.063 | 0.337(0.107~1.059) |
| Type of flap | ALTF | | | 10.765 | 0.056 | |
| | FFF | 1.461 | 0.656 | 4.953 | 0.026* | 4.309(1.190~15.601) |
| | RFFF | -0.444 | 0.468 | 0.899 | 0.343 | 0.641(0.256~1.606) |
| | LDFF | 0.952 | 0.532 | 3.208 | 0.073 | 2.591(0.914~7.345) |
| | LAFF | -0.669 | 1.052 | 0.405 | 0.525 | 0.512(0.065~4.028) |
| | other | 0.897 | 0.650 | 1.903 | 0.168 | 2.453(0.686~8.777) |
| Constant | | -2.88 | 2.872 | 1.006 | 0.316 | 0.056 |

**Alb**:Albumin; **Hb**: Hemoglobin; **PT**:Prothrombin time; **ALTF**:Anterolateral thigh flap; **FFF**:Forehead Flap; **RFFF**:Radial Forearm Free Flap; **LDFF**:Latissimus Dorsi Free Flap; **LAFF:**Lateral Arm Free Flap

Note

*express p<0.05

** express p<0.01

*** express p<0.001.

## 3.3 Multiple logistic regression analysis

Based on the results of the correlation analysis, it is evident that over time, patients' preoperative and postoperative hemodynamics become more stable, leading to a reduced incidence of vascular crises and higher tissue flap survival rates. While these observations could suggest that temporal factors such as improved techniques and experience over the years may influence outcomes, including the year of surgery as a variable in the model could limit its generalizability across different hospitals and regions. To maintain clinical relevance and ensure the broader applicability of the model, we incorporated variables more directly related to surgical outcomes, such as tissue flap type and defect location. Year-related variables were sequentially included, observing the concordance index (C-index) and calibration curve. Following multivariate logistic regression analysis (Table 3), the final model was determined as follows:

LogitP = -2.880+1.061 * Tissue flap width -2.149 * D-dimer+0.013 * preoperative hemoglobin +0.026 * preoperative and postoperative hemoglobin difference+(-0.783 * stapler anastomoses blood vessels/-1.494 * stapler anastomoses and manually sutures blood vessels) - 0.113 *

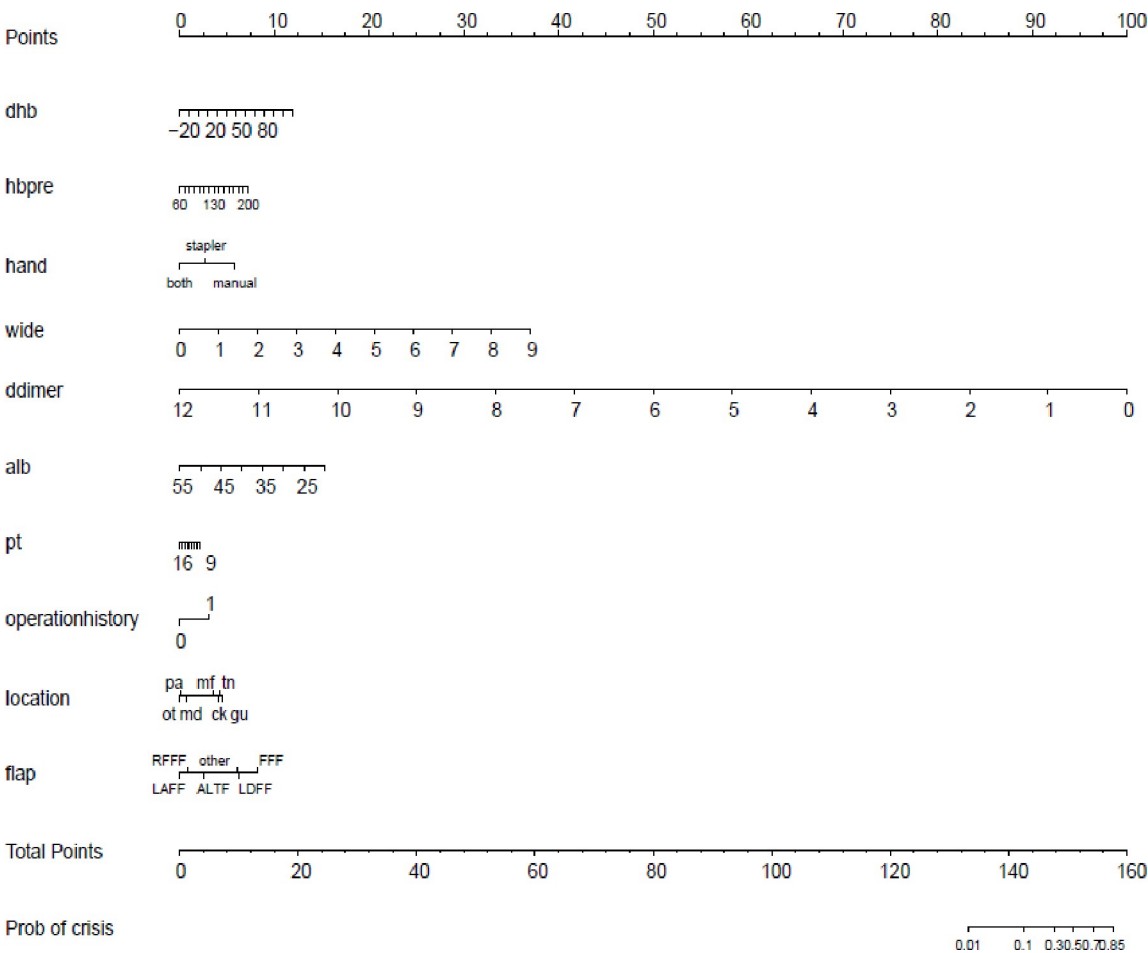

**Fig 1. Nomogram model for predicting vascular crises after free flap repair of oral and maxillofacial tumors.** Nomogram for predicting vascular crises following free tissue flap repair of oral and maxillofacial tumors. The model incorporates key predictors and displays a favorable fit, achieving a cutoff value of 0.069 and explaining 16.9% of the total variation. Internal and external validation results support its effectiveness in clinical applications. **Note:** ot:express other; pa:express palate; md:express mandible; mf: express mouth floor; ck: express cheek; tn: express tongue; gu: express upper gum.

preoperative albumin value—0.067 * prothrombin time+0.804 * Tissue flap surgery history +(-0.016 * cheek/-0.173 * floor of mouth/-0.904 * mandible/+0.089 * maxillary gingiva/-1.054 * palate/-1.089 * other parts)+ (1.461 * Free fibular flap/-0.444 * Free radial forearm tissue flap/+0.952 * Free latissimus dorsi tissue flap/-0.669 * Free lateral upper arm tissue flap/+0.897 * Other types of tissue flap).

### 3.4 Internal and external validation of the nomogram model

The nomogram model (Fig 1) demonstrated effective internal and external validation in predicting vascular crises following free tissue flap repair of oral and maxillofacial tumors. The model exhibited a favorable fit with a cutoff value of 0.069 and a non-significant p-value of 0.424 from the Hosmer-Lemeshow test, indicating strong fitting capability and explaining 16.9% of the total variation. In the training group, the sensitivity, specificity, and AUC were 0.708, 0.764, and 0.780, respectively, with the calibration curve closely aligning with the 45° reference line (Fig 2) Internal validation using the Bootstrap method (1000 samples) showed

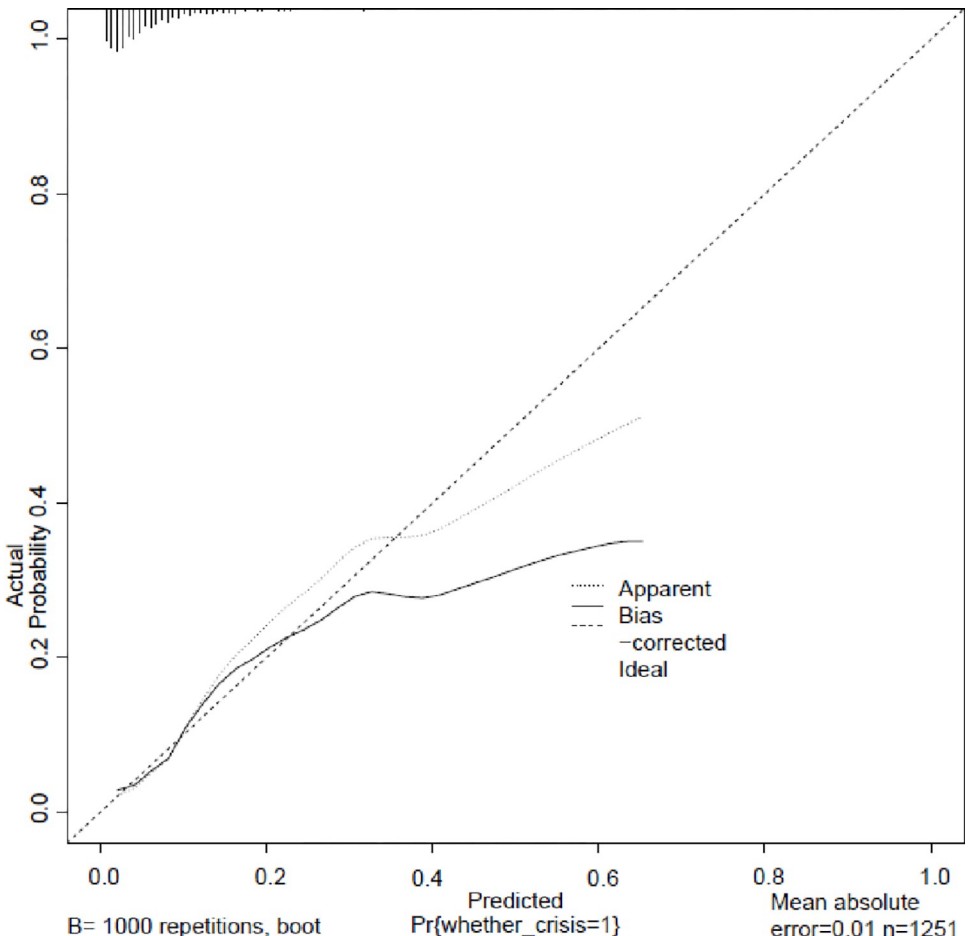

**Fig 2. Calibration curve for internal validation in the training group.** The curve demonstrates a close alignment with the 45° reference line, indicating effective predictive accuracy and calibration of the model for vascular crises following free tissue flap repair of oral and maxillofacial tumors.

an average absolute error of 0.01 for the calibration curve, confirming excellent fit against the ideal curve (Fig 3). The ROC curve of the training group is shown in Fig 4. The validation group exhibited a sensitivity of 0.990 and specificity of 0.038, with an AUC of 0.701 (Fig 5) and a calibration curve resembling that of the training group, further affirming the model's robust performance (Fig 6).

## 3.5 Decision Curve Analysis (DCA)

Decision Curve Analysis (DCA) was conducted to evaluate the clinical applicability of the predictive model. The DCA results demonstrated that the nomogram provided net benefits across a range of threshold probabilities in both the training and validation groups. Specifically, the DCA curve for the training group (Fig 7) showed that the model offers significant clinical value compared to either the "treat-all" or "treat-none" strategies, especially within the risk threshold range of 0.000–0.670. Similarly, the DCA curve for the validation group (Fig 8) confirmed the model's potential to guide clinical decision-making, providing net benefits over the same range of threshold probabilities.

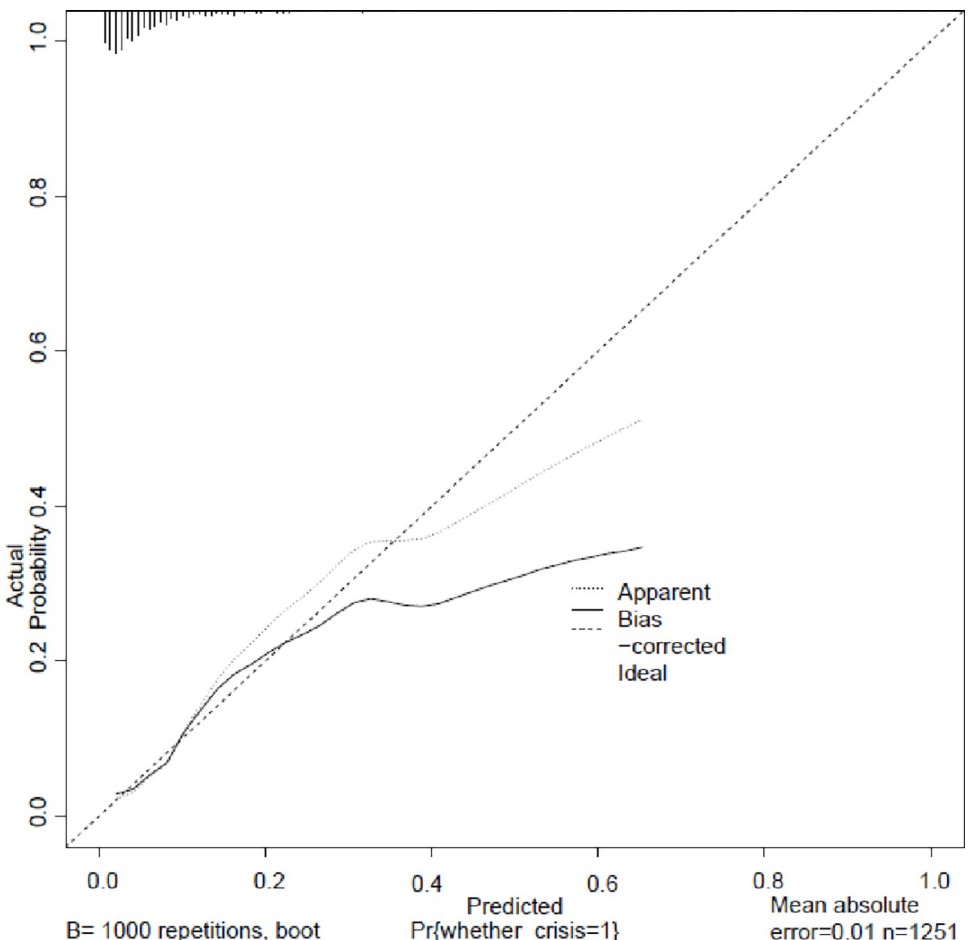

**Fig 3. Training group internal validation.** Calibration curve for internal validation using the Bootstrap method (1000 samples). The average absolute error of 0.01 demonstrates excellent fit against the ideal curve.

## 4. Discussion

In this study, we developed and validated a predictive model for vascular crises following free flap surgery in patients with oral and maxillofacial tumors, employing a prospective, nested case-control study design. Our findings demonstrate the model's robust discriminatory ability and accurate calibration in predicting postoperative vascular crises, highlighting its potential as an objective clinical tool for risk assessment and mitigation in this patient population.

Tumor site and size emerged as significant risk factors for vascular crisis. Larger tumors, associated with extensive wound surfaces and greater vascular damage, necessitate higher blood flow dynamics for tissue flap perfusion. Specifically, defects in the upper gingiva, tongue, and buccal regions were most susceptible to vascular crisis, aligning with previous research findings [21] that postoperative complications increased by 2% for each unit increase in oral and maxillofacial defect volume. Tumor characteristics, including location, size, and thickness, are closely linked to local recurrence and distant metastasis [22].

Our analysis identified previous tissue flap surgeries and the method of arterial anastomosis as significant treatment-related predictors of vascular crisis. The selection of tissue flap critically impacts the choice of donor area blood vessels, particularly with regard to vascular complications. Latissimus dorsi and free fibula flaps are associated with a heightened risk of

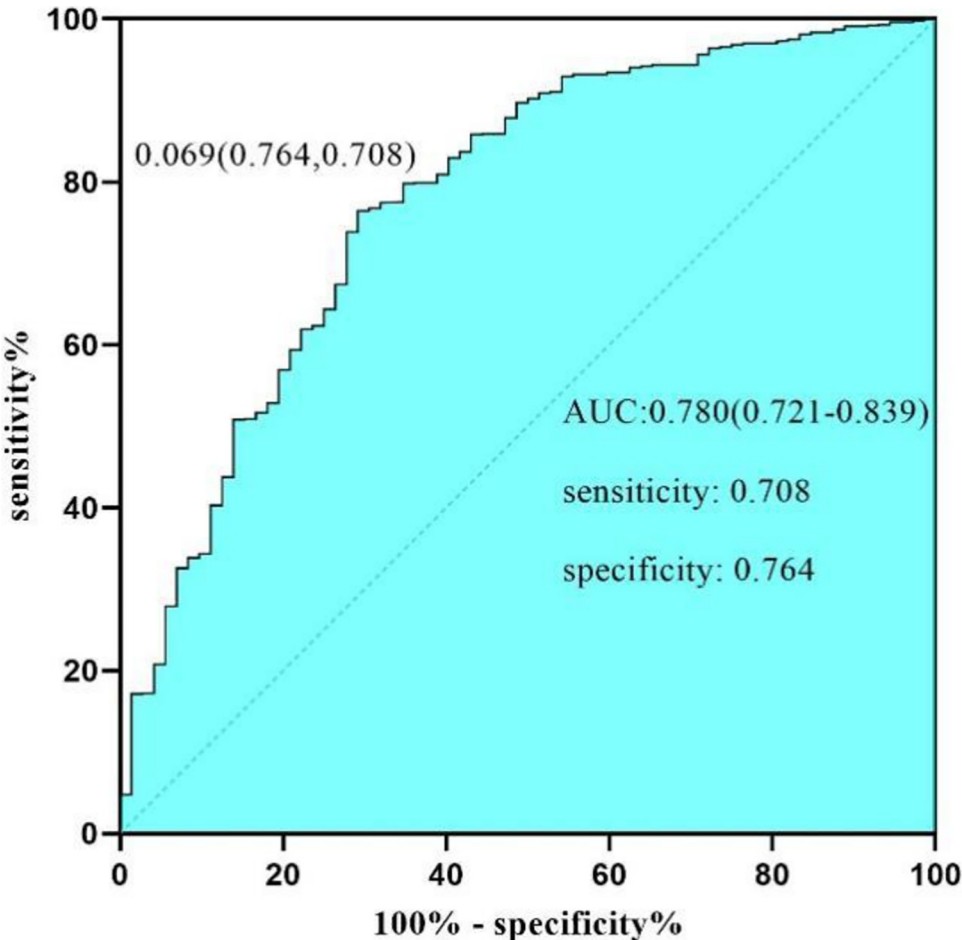

**Fig 4. ROC curve of training group.** ROC curve for the training group, demonstrating the nomogram model's predictive performance for vascular crises after free tissue flap repair in oral and maxillofacial tumors. The AUC is 0.780, with sensitivity of 0.708 and specificity of 0.764, indicating good overall accuracy. The calibration curve aligns closely with the ideal reference line, confirming the model's strong fit.

vascular crisis compared to anterolateral thigh flaps, potentially attributed to their larger defect repairs and greater blood flow requirements [23, 24]. 43 evious tissue flap surgeries may increase the risk of vascular crisis by damaging and shortening recipient area blood vessels, complicating anastomosis, and risking vascular endothelial injury, as noted in prior research [25, 26]. Conversely, stapler-assisted arterial anastomosis reduces the risk of vascular crisis by minimizing vascular trauma and expediting the anastomosis process [27]. Healthcare staff should implement a standardized monitoring protocol that includes hourly assessments of skin color, temperature, flap tension, capillary refill time, and bleeding characteristics for the first 48 hours post-surgery.

Our study revealed a significant correlation between the difference in preoperative and postoperative hemoglobin levels and the incidence of vascular crisis. This relationship highlights the impact of hemodynamic changes in tissue flap vessels, especially in cases where high preoperative red blood cell counts increase blood viscosity and resistance, compromising microcirculatory perfusion and oxygen delivery. Consistent with previous research [28], elevated red blood cell counts were associated with alterations in the fibrinolytic system, potentially exacerbating postoperative microcirculatory dysfunction [29]. To mitigate these risks,

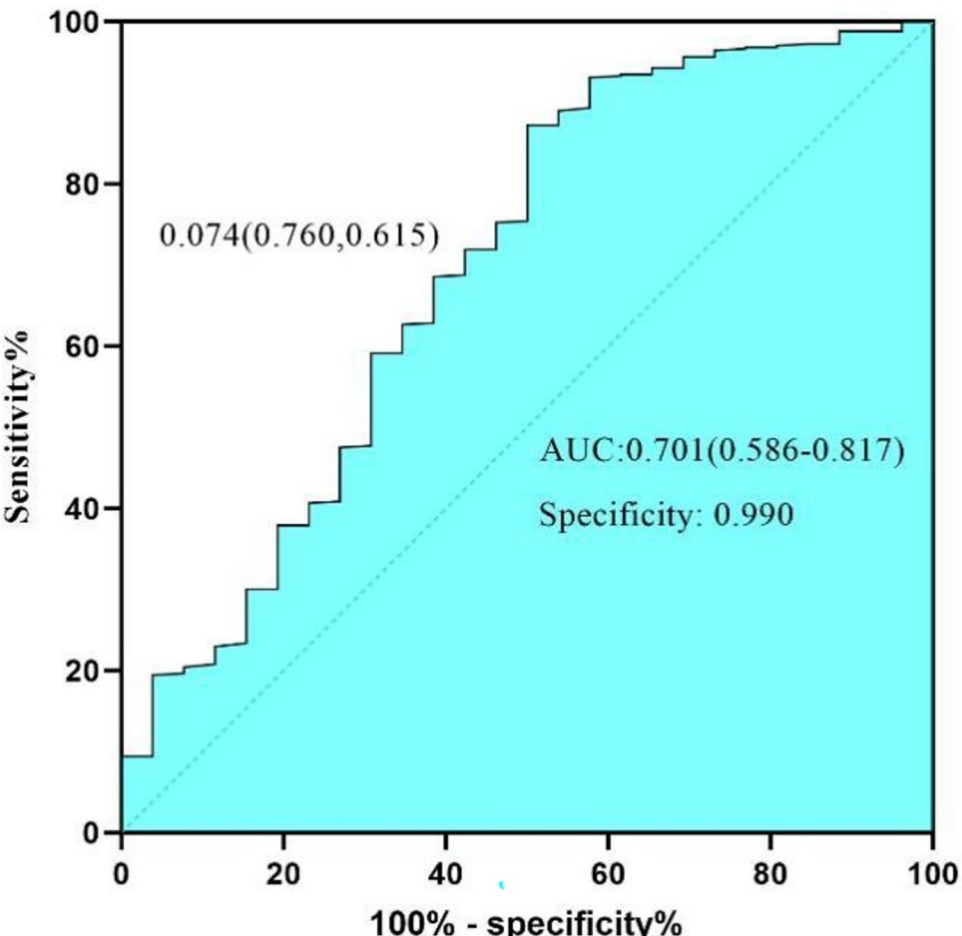

**Fig 5. ROC curve of validation group.** ROC curve for the validation group, illustrating the model's performance in predicting vascular crises post-free tissue flap repair. The AUC is 0.701 (95% CI: 0.586–0.817), indicating moderate predictive accuracy, with a specificity of 0.990 and sensitivity of 0.074. The calibration curve aligns closely with the ideal reference, reinforcing the nomogram's robust external validation.

strategies such as preoperative acute isovolemic hemodilution are recommended to optimize blood flow dynamics and enhance tissue oxygenation [30]. Additionally, low preoperative D-dimer levels were identified as a risk factor, reflecting varying clotting responses. These findings underscore the need for tailored perioperative strategies to optimize hemodynamics and coagulation management, thereby enhancing surgical outcomes [31, 32].

Yu Zhao et al. [33] developed a nomogram model aimed at predicting short-term complications following local flap resection, encompassing various outcomes such as postoperative infection, dehiscence, bleeding, subcutaneous fluid accumulation, fat liquefaction, arteriovenous crisis, among others. However, the model lacks specificity specifically for vascular crises. In contrast, Lese I [14] in 2021 proposed risk prediction indexes stratified into low, moderate, and high categories based on factors including defect cause and comorbidities like coronary heart disease, diabetes, smoking, peripheral arterial disease, and arterial hypertension. Patients with a moderate risk index were found to have a 9.3 times higher likelihood of vascular damage compared to the low-risk group, while those with a high-risk index had an 18.6 times higher likelihood. Nevertheless, Lese I's model extends beyond oral and maxillofacial regions to include limbs and breasts, which diminishes its specificity. Moreover, the model lacks rigorous

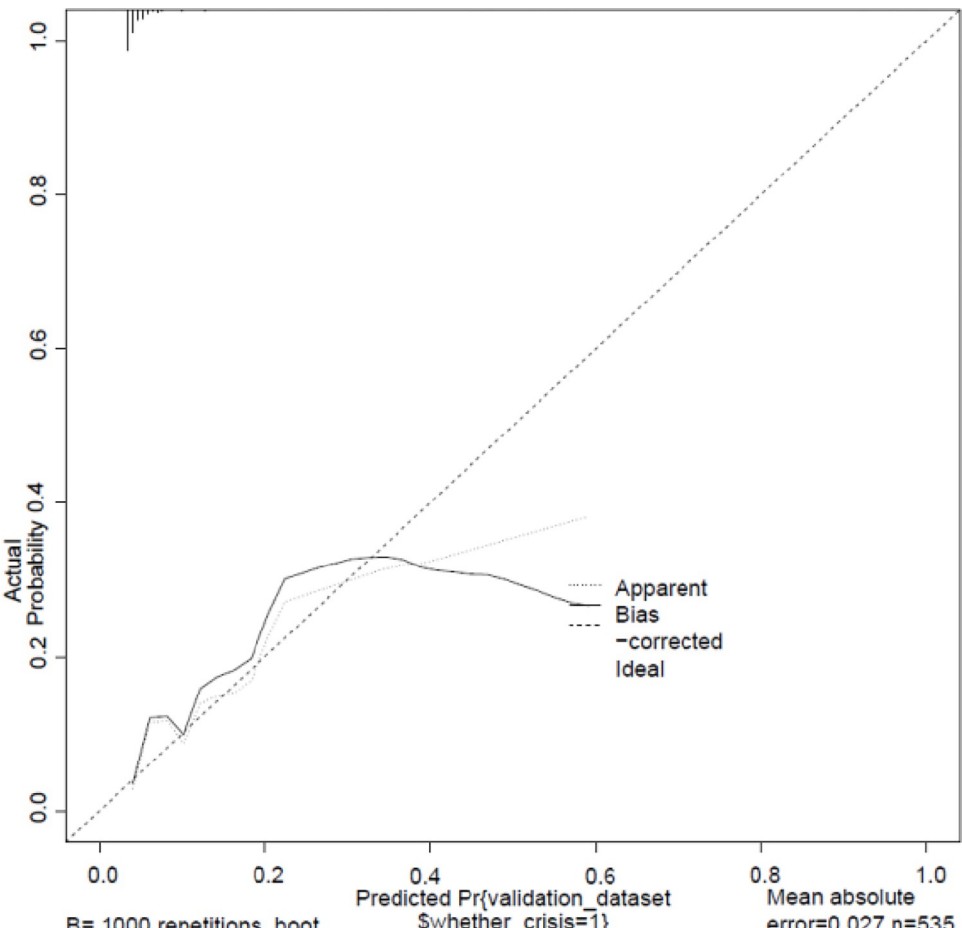

**Fig 6. Calibration curve for external validation in the validation group.** Calibration curve for the validation group. The curve closely resembles that of the training group, affirming the model's robustness and predictive accuracy.

validation, both internally and externally, and requires calibration and discrimination analysis to establish scientific validity. Our study integrated 10 variables to construct a risk prediction model specifically targeting vascular crises. This model demonstrates high specificity and holds significant clinical utility in preemptively ruling out vascular crises, potentially averting unnecessary secondary exploratory surgeries.

Our findings highlight several strategies for optimizing microsurgery outcomes. These include selecting vessels with larger diameters, ensuring adequate anesthesia and analgesia to prevent vascular constriction, and implementing appropriate perioperative fluid rehydration and insulation measures to maintain anastomotic vessel diameters and blood flow. Safeguarding endothelial cell anticoagulation and spasmolytic function while minimizing collagen fiber exposure is crucial to reducing intravascular thrombosis risks [34]. Clinically, our model emphasizes monitoring skin color, temperature, flap tension, capillary refill time, and bleeding characteristics of free tissue flaps. Based on the model's risk assessment, streamlined ICU utilization for low-risk vascular crisis patients can optimize resource allocation, enhance nursing efficiency, reduce costs, and alleviate caregiver burden.

This study acknowledges several limitations. Firstly, the model was developed and validated using data solely from a single institution, potentially restricting the generalizability of findings

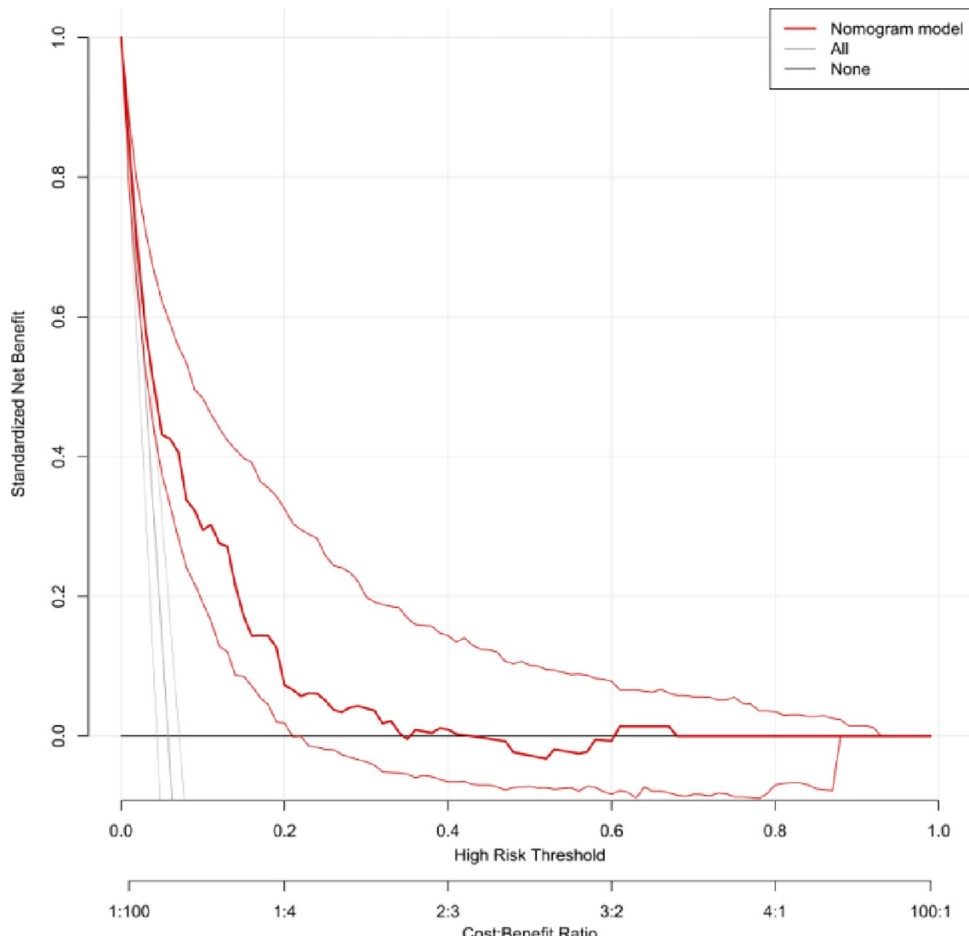

**Fig 7. DCA curve of training group.** The DCA curve illustrates the net benefit of the nomogram model compared to the "treat-all" and "treat-none" strategies across a range of threshold probabilities (0.000–0.670).

to other populations or settings. Secondly, while our model integrated 10 clinically relevant variables, other significant predictors of vascular crises may exist that were omitted or unavailable in our dataset. Finally, although the model demonstrates high specificity and accuracy, its performance necessitates validation across larger and more diverse patient cohorts to substantiate its robustness and reliability. These limitations emphasize the ongoing need for continued research and iterative refinement of the model to maximize its clinical utility and effectiveness.

## 5. Conclusion

In this study, we developed and validated a risk prediction model for vascular crises among patients undergoing free flap surgery for horal and maxillofacial tumors. The model incorporates 10 clinically relevant variables and demonstrates high specificity and accuracy, enabling early identification of risks and targeted interventions. Implementing this model in clinical practice has the potential to optimize resource allocation, improve patient outcomes, and reduce unnecessary surgeries, thereby enhancing overall care efficiency.

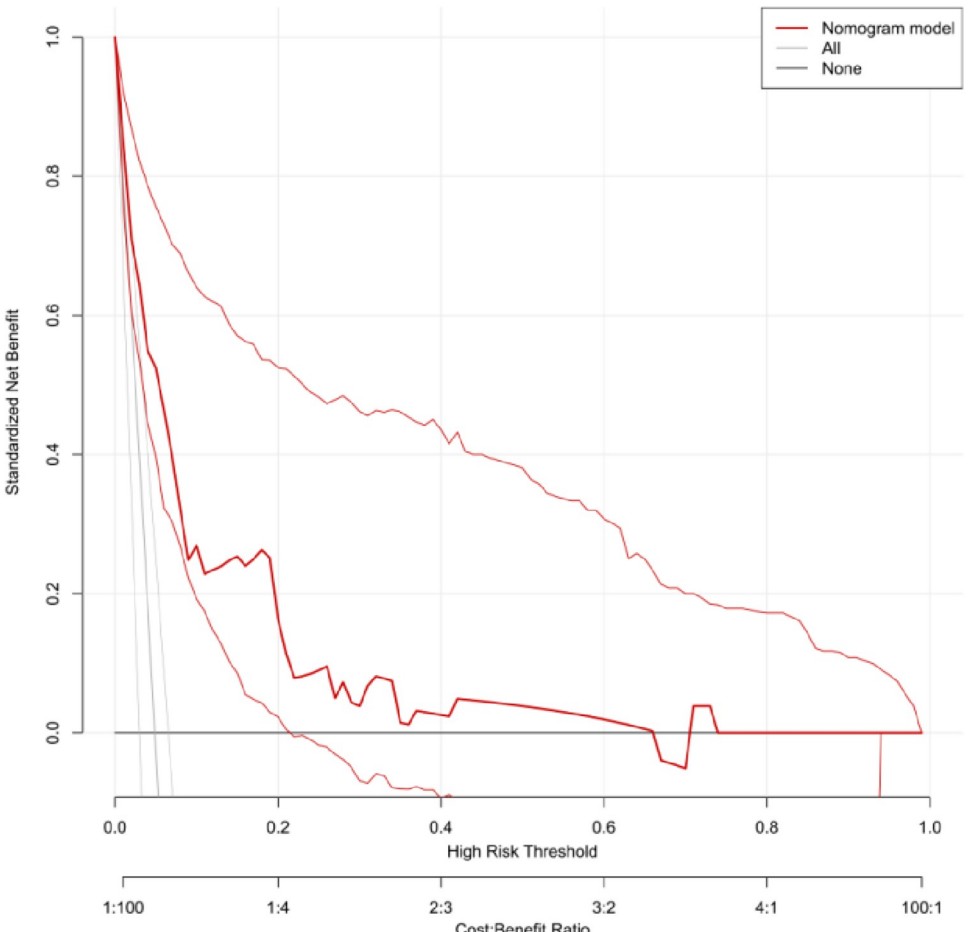

**Fig 8. DCA curve of validation group.** The DCA curve confirms the model's clinical applicability, showing net benefits over a similar range of threshold probabilities.

## Acknowledgments

The authors extend appreciation to the Department of Head and Neck Oncology staff, West China Hospital of Stomatology, Sichuan University.

## Author Contributions

**Conceptualization:** Xiaoqin Bi.

**Data curation:** Ying Zheng, Jingya Yu.

**Formal analysis:** Ying Zheng, Yunyu Zhou, Qian Lu.

**Funding acquisition:** Xiaoqin Bi.

**Methodology:** Ying Zheng.

**Software:** Jingya Yu, Yunyu Zhou, Qian Lu, Yu Zhang.

**Supervision:** Jingya Yu, Yu Zhang, Xiaoqin Bi.

**Writing – original draft:** Ying Zheng.

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
