## [Decision Letter · Decision Letter 0]

17 Sep 2024

PONE-D-24-33122Development and Validation of a Predictive Nomogram for Vascular Crises in Oral and Maxillofacial Cancer Patients Undergoing Free Flap SurgeryPLOS ONE

Dear Dr. Bi,

Thank you for submitting your manuscript to PLOS ONE. After careful consideration, we feel that it has merit but does not fully meet PLOS ONE’s publication criteria as it currently stands. Therefore, we invite you to submit a revised version of the manuscript that addresses the points raised during the review process.

We look forward to receiving your revised manuscript.

Kind regards,

Stanisław Jacek Wroński, M.D., Ph.D, FEBU

Academic Editor

PLOS ONE

Journal Requirements:

4. Thank you for stating the following financial disclosure: [This work was supported by the Sichuan Science & Technology Project [Grant number 2022JDKP007] and Medical Research Project of Chengdu Health Commission[Grant number 2022015]. Medical research topic in Sichuan Province[Grant number S23099].]. Please state what role the funders took in the study. If the funders had no role, please state: "The funders had no role in study design, data collection and analysis, decision to publish, or preparation of the manuscript." If this statement is not correct you must amend it as needed. Please include this amended Role of Funder statement in your cover letter; we will change the online submission form on your behalf.

5. Please provide a complete Data Availability Statement in the submission form, ensuring you include all necessary access information or a reason for why you are unable to make your data freely accessible. If your research concerns only data provided within your submission, please write "All data are in the manuscript and/or supporting information files" as your Data Availability Statement.

6. PLOS requires an ORCID iD for the corresponding author in Editorial Manager on papers submitted after December 6th, 2016. Please ensure that you have an ORCID iD and that it is validated in Editorial Manager. To do this, go to ‘Update my Information’ (in the upper left-hand corner of the main menu), and click on the Fetch/Validate link next to the ORCID field. This will take you to the ORCID site and allow you to create a new iD or authenticate a pre-existing iD in Editorial Manager.

7. Please include a separate caption for each figure in your manuscript.

Additional Editor Comments:

Dear Authors,

submitted manuscript "Development and Validation of a Predictive Nomogram for Vascular Crises in Oral and Maxillofacial Cancer Patients Undergoing Free Flap Surgery" PONE-D-24-33122, noode some major revisions. Please find the following comments:

1. The manuscript must describe a technically sound piece of scientific research with data that supports the conclusions. Experiments must have been conducted rigorously, with appropriate controls, replication, and sample sizes. The conclusions must be drawn appropriately based on the data presented.Is the manuscript technically sound, and do the data support the conclusions? Answer: Partly.

2.This interesting paper shows a large study about a very wide series of free-flaps performed for face reconstruction after cancer, with an interesting "prospectiv in the past" pattern

In my opinion, some major comments can be addressed :

1) description of the cohort is incomplete. Data such as radiation therapy, type of free flap per patient should be provided

2) these data should be used in the comparison between training and validation cohorts

3) it isn't clear whether the biological data are pre or post operative ones, therefore if patients can be classified in "high risk" and "low risk" groups PRE operatively, which would be at greater value

4) association between vascular crisis and year of surgery is not clear : is it supposed to depend on surgeon's experience ?

5) the nomogram, which is the major result of this sudy, has to be publicly available

6) I think that a specific statistical reviewing is mandatory, to validate this intersting tool

As minor comment, the autors should specify which package of R software they used

As a general impression, an interesting tool, a huge cohort, but specifications of the cohort and discussion both need more precision

with compliments

Stanisław Wroński

Academic Editor

Reviewers' comments:

Reviewer's Responses to Questions

**Comments to the Author**

1. Is the manuscript technically sound, and do the data support the conclusions?

Reviewer #1: Partly

2. Has the statistical analysis been performed appropriately and rigorously? 

Reviewer #1: I Don't Know

3. Have the authors made all data underlying the findings in their manuscript fully available?

Reviewer #1: Yes

4. Is the manuscript presented in an intelligible fashion and written in standard English?

Reviewer #1: Yes

5. Review Comments to the Author

Reviewer #1: This interesting paper shows a large study about a very wide series of free-flaps performed for face reconstruction after cancer, with an interesting "prospectiv in the past" pattern

In my opinion, some major comments can be addressed :

1) description of the cohort is incomplete. Data such as radiation therapy, type of free flap per patient should be provided

2) these data should be used in the comparison between training and validation cohorts

3) it isn't clear whether the biological data are pre or post operative ones, therefore if patients can be classified in "high risk" and "low risk" groups PRE operatively, which would be at greater value

4) association between vascular crisis and year of surgery is not clear : is it supposed to depend on surgeon's experience ?

5) the nomogram, which is the major result of this sudy, has to be publicly available

6) I think that a specific statistical reviewing is mandatory, to validate this intersting tool

As minor comment, the autors should specify which package of R software they used

As a general impression, an interesting tool, a huge cohort, but specifications of the cohort and discussion both need more precision

6. PLOS authors have the option to publish the peer review history of their article (what does this mean?). If published, this will include your full peer review and any attached files.

Reviewer #1: No

---

## [Author Response · Author response to Decision Letter 0]

24 Sep 2024

Thank you to the editor and reviewers for their valuable feedback and suggestions. We have carefully addressed all comments and detailed the specific modifications in the uploaded "Response to Reviewers" file. We believe these changes have significantly improved the quality of our manuscript. Thank you for your support and guidance!

---

## [Decision Letter · Decision Letter 1]

15 Nov 2024

Development and Validation of a Predictive Nomogram for Vascular Crises in Oral and Maxillofacial Cancer Patients Undergoing Free Flap Surgery

PONE-D-24-33122R1

Dear Dr. Xiaoqin Bi

We’re pleased to inform you that your manuscript has been judged scientifically suitable for publication and will be formally accepted for publication once it meets all outstanding technical requirements.

Kind regards,

Stanisław Jacek Wroński, M.D., Ph.D, FEBU

Academic Editor

PLOS ONE

Additional Editor Comments

Dear Authors,

after careful consideration of revised version and the reviewer's opinion, I conclude that in its present form the article can be approved for publication in POLS ONE.

The authors have effectively addressed all concerns raised, resulting in a significantly improved manuscript.

With compliments

Stanisław Wroński

Academic Editor

Reviewers' comments:

Reviewer #1: All comments have been addressed. Thank you for your precise answers to my questions

Impressive series indeed ! the case mix, with a majority of ALT, is quite uncommon for max-fac, as we genrally use FFF and ABF as main flaps in facial reconstruction

---

## [Editor Report · Acceptance letter]

21 Nov 2024

PONE-D-24-33122R1 

PLOS ONE

Dear Dr. Bi, 

I'm pleased to inform you that your manuscript has been deemed suitable for publication in PLOS ONE. Congratulations! Your manuscript is now being handed over to our production team.

Kind regards, 

on behalf of

Dr. Stanisław Jacek Wroński 

Academic Editor

PLOS ONE